# Molecular Epidemiology of Human Adenovirus from Acute Gastroenteritis Cases in Brazil After the COVID-19 Pandemic Period, 2021–2023

**DOI:** 10.3390/v17040577

**Published:** 2025-04-17

**Authors:** Mateus de Souza Mello, Fábio Correia Malta, Alexandre Madi Fialho, Fernanda Marcicano Burlandy, Tulio Machado Fumian

**Affiliations:** Laboratório de Virologia Comparada e Ambiental, Instituto Oswaldo Cruz (IOC), Fundação Oswaldo Cruz (FIOCRUZ), Rio de Janeiro CEP 21045-900, Brazil; mello.souza.mateus@gmail.com (M.d.S.M.); fabio.malta@ioc.fiocruz.br (F.C.M.); amfialho@ioc.fiocruz.br (A.M.F.); fburlandy@ioc.fiocruz.br (F.M.B.)

**Keywords:** human adenovirus (HAdV), acute gastroenteritis (AGE), molecular epidemiology, HAdV-F40/41

## Abstract

Human enteric adenoviruses (HAdV-F40/41) play a crucial role as causative agents of acute gastroenteritis (AGE), particularly affecting children in low-and middle-income countries. This study investigated the prevalence, genetic diversity, and molecular characteristics of HAdV-F40/41 in AGE cases reported in Brazil from 2021 to 2023, a period after the COVID-19 pandemic. A total of 1980 stool samples collected from medically attended AGE patients from nine states were analyzed by TaqMan-based qPCR. Overall, HAdV was detected in 16.6% (*n* = 328/1980) of cases, with the highest prevalence observed in children under five years of age. The positive HAdV samples were genotyped through partial sequencing of the hexon and/or fiber genes followed by phylogenetic analysis. Enteric HAdVs (HAdV-F40/41) were detected in 3.2% (*n* = 63/1980) of samples, with HAdV-F41 (44.1%) being the most common genotype. Among the non-enteric types, HAdV-C (29.4%) was the most prevalent, followed by HAdV-B (13.2%), HAdV-A (10.3%), and HAdV-D (2.9%). Phylogenetic analysis of the hexon (HVR1–HVR6) and fiber (Shaft) gene regions identified two major clusters, H-GTC1 and F-GTC2, showing close genetic relationships with global strains. HAdV-F40/41 demonstrated significantly higher viral loads compared to non-enteric HAdVs. These findings highlight the importance of continued surveillance of HAdV-F to better understand its role in AGE cases and support public health strategies, including potential vaccine development.

## 1. Introduction

Acute gastroenteritis (AGE) is a leading cause of death worldwide, particularly in low-and middle-income countries (LMICs) [1,2]. It accounts for approximately 1.7 billion cases and 1.1 million deaths annually worldwide, primarily affecting children under 5 years of age in LMICs [3,4]. Among the viral agents related to AGE, the most common viruses involved in epidemic cases and outbreaks include rotavirus, norovirus, human adenovirus (HAdV), astrovirus, and sapovirus [5,6]. According to the International Committee on the Taxonomy of Viruses (ICTV), HAdVs belong to the *Adenoviridae* family, within the *Mastadenovirus* genus (ICTV; https://ictv.global/, accessed on 15 January 2025) [7]. They are non-enveloped viruses that comprise a linear double-stranded DNA genome measuring approximately 35 kb in size, containing multiple open reading frames (ORFs) that encode proteins with different functions [8,9,10]. The structure of the viral capsid is icosahedral and can be classified into three distinct groups: major capsid proteins (hexon, penton base, and fiber), minor proteins (IIIa, VI, VIII, and IX) and non-structural proteins (IVa2, V, VII, μ, terminal, and protease) [11].

HAdVs are divided into seven different species (A–G), and 116 distinct genotypes have been identified using molecular methodologies (http://hadvwg.gmu.edu/, accessed on 15 January 2025) [12,13]. The symptomatic profile of each HAdV varies depending mainly on intrinsic factors (tropism and receptor affinity) such as viral extrinsic factors (age and immunodeficiency of the individual) [14,15,16,17,18]. These viruses can be responsible for manifestations associated with the respiratory, gastrointestinal, and ocular tracts, in addition to rare infections of the urinary tract, hepatocytes, and nerve cells [19]. While immunocompetent individuals tend to develop a self-limiting condition, immunosuppression patients, such as transplant recipients, tend to have a more complex development of the pathology [20,21,22]. Genotypes 40 and 41, within the F species (also known as enteric adenoviruses), have been well recognized as important cause of AGE, primarily affecting children under 5 years of age and accounting for around 3% to 12% of infantile AGE cases [23,24,25,26,27].

Cohen et al. [28], who have worked in LMICs, have identified HAdV-F40/41 as the second most frequently detected pathogen in children under five years of age with AGE, behind only rotaviruses [28]. A study on HAdV-F41 in children younger than six years old has identified distinct mutation profiles, with the short fiber region demonstrating greater permissiveness to these genetic changes [29]. A 10-year study conducted in Kenya investigating HAdV genetic diversity in children under 13 years old identified five HAdV-F40 and seven HAdV-F41 clades, highlighting previously unrecognized viral heterogeneity in the region [30]. A recent study performed in England during the COVID-19 pandemic demonstrated a difference in the prevalence of lineages and the predominance of the 2b lineage [31].

In Brazil, the rotavirus vaccine (RV1, Rotarix) was incorporated into the National Immunization Program in 2006, resulting in a significant reduction in RVA-related infections, hospitalizations, and deaths [32]. In previous studies from our group, we demonstrated an increase in noroviruses and HAdV circulation post-rotavirus vaccine introduction, showing these agents as a major cause of AGE in Brazil [33,34]. Few studies have been carried out to elucidate either the epidemiology or genetic diversity of HAdV-F40/41 [35,36,37]. In a previous study conducted by our group, HAdV was detected in 24.5% of AGE cases in Brazil between 2018 and 2020 [33]. Additionally, during the COVID-19 pandemic, cases of severe acute hepatitis of unknown etiology were reported in the United States and Ireland; some cases were linked to HAdV-F41 infections, highlighting the importance of HAdV surveillance at the country level [38,39]. Therefore, our study aimed to investigate the circulation, prevalence, and molecular diversity of HAdV during a three-year period (2021–2023) in AGE cases in Brazil. We also explored the genetic variability of enteric HAdV-F40/41 by sequencing the six hypervariable regions from the hexon gene (HVR1–HVR6), as well as the shaft in the long-fiber region.

## 2. Materials and Methods

### 2.1. Stool Samples and Ethical Aspects

Over three years, between January 2021 and December 2023, this study analyzed 1980 stool samples from medically attended patients (both children and adults) with AGE symptoms from nine states across the Southern, Southeastern, and Northeastern regions of Brazil. AGE was characterized by the sudden onset of diarrhea, with at least three liquid or semi-liquid stools within a 24 h period, with or without fever and vomiting. Stool samples were collected from clinically attended patients with AGE, together with clinical–epidemiological data. Samples were forwarded to the Laboratory of Comparative and Environmental Virology, which houses the Regional Reference Laboratory for Rotavirus (RRRL) at the Oswaldo Cruz Institute, Fiocruz, and were kept frozen at −20 °C until use. The RRRL is part of the national rotavirus surveillance program, under the supervision of the General Coordination of Public Health Laboratories of the Brazilian Ministry of Health.

This study is currently approved by the Ethics Committee of the Oswaldo Cruz Foundation (FIOCRUZ), Brazil (Approval number: CAAE: 76063123.5.0000.5248), and conducted according to the guidelines of the Declaration of Helsinki. Stool samples were manipulated anonymously, and patient data were maintained securely in compliance with the Ethical Protocol Statement ABNT NBR ISO 15189:2015. Patient-informed consent was waived by the Fiocruz Ethical Committee, and patients’ data were maintained anonymously and securely.

### 2.2. Viral Nucleic Acid Extraction

Viral nucleic acids were extracted from 140 μL of clarified stool suspension (10% *w*/*v*) using Tris-calcium buffer (pH = 7.2). Samples underwent an automated nucleic acid extraction process utilizing the QIAcube^®^ automated system and the QIAamp Viral RNA Mini Kit (both from QIAGEN, Valencia, CA, USA), in accordance with the manufacturer’s guidelines. Nucleic acids were eluted using 60 µL of the elution buffer AVE and were promptly stored at −80 °C prior to molecular analysis. In every extraction procedure, RNAse/DNAse-free water served as the negative control.

### 2.3. HAdV Detection and Quantification

A multiplex TaqMan-based quantitative PCR (qPCR) method was employed for detection and quantification of all HAdV types and specifically the enteric F types. The multiplex qPCR combined a degenerate set of primers and a probe targeting a conserved region of the hexon gene to detect all HAdV types, in addition to primers and a probe used specifically for HAdV-F targeting the fiber region [40,41]. Detailed information on initial HAdV detection and quantification methods has been previously described [42]. Briefly, multiplex qPCR reactions were performed with 5 µL of the extracted DNA in a final volume of 20 µL, containing 10 µL of the 2× QuantiTect Probe PCR Kit (Qiagen, Valencia, CA, USA) and primers and probes with final concentrations of 1 µM and 0.25 µM, respectively. Reactions were conducted in the Applied Biosystems 7500 Real-Time PCR System (Applied Biosystems, Foster City, CA, USA) under the following thermal cycling conditions: 2 min at 50 °C, 15 min at 95 °C, 40 cycles of 15 s at 95 °C, and 1 min at 60 °C. Samples that exhibited a characteristic sigmoid curve and crossed the threshold line with a cycle threshold (Ct) value < 40 were considered positive. All runs included negative and positive controls (stool sample), as well as a non-template control. This study used all the rotavirus-and norovirus-negative samples (*n* = 1590), and additionally, HAdV was tested in ~35% of the positive samples for rotavirus and norovirus (*n* = 390).

### 2.4. Rotavirus and Norovirus Detection

Rotavirus and norovirus were identified and quantified using TaqMan-based RT-qPCR protocols. For rotavirus, primers (NSP3F and R) and a probe (NSP3p) were used to amplify the conserved NSP3 gene. For the detection of noroviruses GI and GII, it was used primer pairs (COG1F and R; COG2F and R) and probes (RING1C and RING2) targeted the ORF1/2 junction region, respectively [43,44].

### 2.5. HAdV Molecular Characterization and Genotyping

Conventional PCR was performed for the hexon protein using the primers Ad1 and Ad2, with an expected product of between 605 and 629 bp [45]. The reaction was carried out using 5 µL of extracted viral DNA and the 5× QIAGEN OneStep RT-PCR kit (Qiagen, Valencia, CA, USA), with primers at a final concentration of 0.2 µM. The reaction was performed under the following conditions: 1 holding step of 15 min at 95 °C followed by 40 cycles containing 1 min of denaturation at 95 °C, 1 min of annealing at 50 °C, and 1 min of extension at 72 °C. After 40 cycles, a final extension step at 72 °C was performed for 10 min. The PCR products were stored at −20 °C until the purification step.

For HAdV-F strains, we targeted the six hypervariable regions (HVR1–HVR6) of the hexon gene using the primers S29 and S52 and the partial shaft region of the long fiber gene using the primers AdF1 and AdF2 [46,47]. The PCR reactions were performed using the Platinum Taq DNA Polymerase enzyme (Invitrogen, Carlsbad, CA, USA), with 5 μL of extracted DNA in a final reaction volume of 25 μL. The expected amplicons for HAdV-F types 40 and 41 were 640 and 664 nt for the hexon gene and 508 and 530 nt for the fiber gene. All the amplicons were purified using the QIAquick Gel Extraction Kit (Qiagen, Hilden, Germany), following the manufacturer’s recommendations. Sequencing reactions of the purified amplicons were performed using the Big Dye Terminator v.3.1 Cycle Sequencing Ready Reaction Kit on an ABI Prism 3730xl Genetic Analyzer (Applied Biosystems, Foster City, CA, USA) at the Fiocruz Institutional Genomic Platform for DNA sequencing (PDTIS).

### 2.6. Phylogenetic Analysis

HAdV chromatogram and consensus sequence analyses were performed using Geneious Prime 2021.1.1 (Biomatters Ltd., Auckland, New Zealand). For HAdV species and type assignment, an initial sequence analysis was performed using the Basic Local Alignment Search Tool (BLAST, accessed on 5 October 2024). HAdV reference sequences were obtained from the National Center for Biotechnology Information (NCBI) GenBank database and its outgroup; all sequences were aligned using MAFFT [48,49]. Phylogenetic trees of the partial hexon gene were constructed using the maximum likelihood method and selected the best-fit evolutionary model from jModeltest 2.1.6 for the dataset using the Tamura-Nei (TN93) +G + I model in MEGA X v. 10.2.6 [50,51,52]. The phylogenetic reconstruction method used was the maximum likelihood (1000 bootstraps iterations). Reference sequences were obtained from the National Center for Biotechnology Information (NCBI) GenBank database. In addition, to investigate the genetic diversity of the Brazilian HAdV-F strains, synonymous and non-synonymous mutations in the sequenced portions of the hexon and fiber genes were compared for each gene with prototype strains obtained from the GenBank database.

HAdV-F40/41 reference sequences were obtained from the National Center for Biotechnology Information (NCBI) GenBank database and its outgroup, and all sequences were aligned using MAFFT [48,49]. Phylogenetic trees of the partial hexon and fiber gene were constructed using the maximum likelihood method and the best-fit evolutionary model from jmodeltest 2.1.6 was selected for the dataset using the Hasegawa–Kishino-Yano (HKY) model for the fiber gene and HKY + I for the hexon gene [50,52,53]. The phylogenetic reconstruction method used was the maximum likelihood (1000 bootstraps iterations). The nucleotide sequences obtained in this study were deposited in the GenBank database under the accession numbers PV344519-PV344574 and PV392539-PV392576.

### 2.7. Statistical Analysis

Statistical analyses were performed using GraphPad Prism software v.9.0.0 (GraphPad Software, San Diego, CA, USA). Mann–Whitney U tests were used to assess significant differences between HAdV detection rates, years of collecting samples, viral load values between single and co-detection, age groups, and types. Fisher’s exact tests were used for analyzing categorical characteristics in contingency tables. For all analyses, a *p*-value  ≤  0.05 was statistically significant.

## 3. Results

### 3.1. Epidemiologic Features of HAdV

Between January 2021 and December 2023, a total of 1980 stool samples were evaluated, both from patients with negative results for rotavirus and norovirus (*n* = 1590) and from samples positive for those viruses (*n* = 390). Overall, we detected HAdV in 16.6% (328/1980) of samples. Among the rotavirus- and norovirus-negative and positive samples, HAdV was detected in 15.7% and 20.4%, respectively. Enteric types HAdV-F40/41 were detected in 3.2% (*n* = 63) of the stool samples.

HAdV detection rates varied from 13% (76/583) in 2021 to 20.1% (145/723) in 2023 (Table 1). An increasing detection of HAdV was observed over the three years, with a significantly higher prevalence observed in 2023 compared to 2021 (*p* = 0.0008) and 2022 (*p* = 0.0437). During the three-year period, HAdV was detected in almost all months, except April and June 2021. Monthly detection rates varied from 4.6% in July 2022 to 31.4% in December 2023, with the latter being the highest detection rate observed during the study period (Figure 1A).

Over the cumulative period from 2021 to 2023, HAdV prevalence ranged from 12.6% in January to 32.9% in December (Figure 1B). With regards to seasonality, the prevalence was higher in spring (17.8%) compared to fall (15.4%), winter (15.9%), and summer (16%), although no significant differences were observed between seasons. (Figure 1C). During surveillance, the northeast revealed a high prevalence of HAdV (20.4%) when compared to the southeast (18.8%) and the south (14.6%), of which the detection in the south was significantly lower than that observed for the other two regions (Table 2). In the southeastern region, HAdV detection rates in 2023 were significantly higher compared to both 2021 (*p* = 0.0002) and 2022 (*p* ≤ 0.0001). The detection rates per year are shown in Table 2.

The frequency of HAdV detection varied across age groups, ranging from 6.9% (57/823) in patients over 60 months to 24.7% (93/337) in children aged 12 to 24 months, and the detection rate was significantly higher among children up to 60 months old compared to older patients (Table 3).

### 3.2. Viral Load and Co-Detection Rates

The viral loads found in the samples ranged from 1.7 × 10^2^ GC/g (genomic copies per gram of feces) to 2.3 × 10^12^ GC/g. When evaluating enteric and non-enteric HAdV, a significantly higher DNA load was observed in stool samples from patients positive for enteric HAdV compared to non-enteric strains (median of 1 × 10^8^ vs. 1.2 × 10^4^ GC/g, respectively) (Figure 2). With regard to age groups, we found a similar DNA viral load across the age groups analyzed, with median values varying from 5.8 × 10^3^ GC/g in the group > 60 months to 6.4 × 10^4^ GC/g in the age group of 24–60 months old.

In order to analyze the rate of co-detection of HAdV with rotavirus and norovirus, we selected around 30% of samples that previously tested positive for both viruses during the study period. Among rotavirus- and norovirus-positive samples, the HAdV detection rate was 24.1% (28/116), and 18.6% (51/274), respectively. Although HAdV was more frequently detected among rotavirus-positive samples, no significant difference was observed between the detection rates (*p* = 0.2719). Interestingly, among HAdV-positive samples that were co-infected with either rotavirus or norovirus, the level of HAdV DNA shedding was significantly lower (*p* ≤ 0.0001) compared to the RNA load of the other viruses. Specifically, the median RNA load for rotavirus was 9.3 × 10^8^ GC/g, and for norovirus it was also 1.4 × 10^10^ GC/g. In contrast, the median DNA load observed for HAdV was markedly lower, at 2.2 × 10^4^ GC/g and 1.5 × 10^5^ GC/g, respectively (Figure 2). Among the 79 HAdV-positive samples co-detected with rotavirus or norovirus, enteric and non-enteric HAdV were detected in seven (1.5%) and 73 (98.5%), respectively (*p* ≤ 0.0001). The higher frequence of non-enteric HAdV in co-detected samples, in relatively low levels, may indicate a less active or secondary role in the infection.

### 3.3. Genetic Characterization HAdV

Among the HAdV analyzed, 20.7% (68/328) of samples collected between 2021 and 2023 were successfully sequenced. The obtained sequences were classified as species A, B, C, D, and F (Figure 3A), with HAdV-F being the most prevalent (44.1%). This species included genotypes F40 and F41, with HAdV-F41 being predominant across most age groups, showing a prevalence varying from 44% to 75% of the samples. In contrast, HAdV-F40 was identified at detection rates between 4% and 10.5%. An exception was observed in children aged between 12 and 24 months, where the prevalence of HAdV-F41 dropped to 15.8%. The second most frequent species was HAdV-C (29.4%), represented by genotypes C1, C2, and C5. Within this group, HAdV-C2 emerged as the dominant genotype in children aged >12–24 months (31.6%). The remaining species (A, B, and D) exhibited lower prevalence rates, ranging from 5% to 25% (Figure 3B). The complete quantitative distribution of HAdV strains is detailed in Figure 3C.

Among the non-enteric HAdV types, the most frequently detected genotype was HAdV-C2 (*n* = 13) (Figure 4). These isolates exhibited high genetic identity (98.3–100%) and were resolved into two distinct clusters with robust statistical support (99% bootstrap), suggesting potential intragenotypic variation. HAdV-B3 was the second most prevalent genotype (*n* = 9), with strains exhibiting 100% nucleotide identity among the Brazilian samples analyzed. The strains were closely related to a Germany strain (OR487155), demonstrating 99.4% of nucleotide identity. HAdV-A31, the third most detected genotype (*n* = 6), exhibited the highest intragenotypic divergence. Notably, one isolate (BRA/LVCA35542/2023/HAdV-31) showed reduced identity (95.5%) compared to others HAdV-A31 strains, highlighting potential evolutionary or geographic distinctions (Figure 4).

### 3.4. Diversity of Enteric HAdV

We characterized partial regions of the hexon and fiber genes from enteric HAdV-F40/41. A total of 28 samples were sequenced for the HVR1-HVR6 hexon region, and 30 samples were sequenced for the shaft of the fiber gene. During the study period, HAdV-F41 was the predominant genotype, detected in 86.7% (*n* = 26) of the samples, while HAdV-F40 was identified in 13.3% (*n* = 4). Regarding HAdV-F40, two samples identified from the fiber region were not represented in the phylogenetic reconstruction.

The phylogenetic analysis of the hexon region (HVR1-HVR6) of the Brazilian HAdV-F41 lineages revealed the presence of two lineages, H-GTC 1 and H-GTC 2 (Figure 5A), of which the majority of samples (*n* = 21) were identified within the H-GTC 1, while six samples were characterized as H-GTC 2. A high nt identity for the Brazilian isolates was observed in Cluster HGTC 1 (99.1–100%), with strains from Brazil (OQ442280 and OQ442288), India (MT952508), France (MW567963), Germany (ON442330), South Africa (MK962806), Japan (LC790284, AB610523 and LC790290), China (PP844976, KY316160, KY316162 and MH465394), and the prototype strain TAK (DQ315364).

A high nt identity (>99% identity) was observed between samples from cluster H-GTC 2 and sequences from the United States (KF303069), Brazil (OQ442277, OQ442282 and OQ442296), and India (MT952501). The nucleotide (nt) identity between sequences HGTC 1 and HGTC 2 was 96.3%.

The HAdV-F40 genotype identified in this study exhibited a high nt identity, varying from 99.3% to 100%, with Brazilian isolates (OQ442264 and OQ442267), the prototype strain Dungan (L19443), and sequences reported from South Africa (MK955319) and India (MT952448). The high genetic conservation between geographically distinct HAdV-F40 strains reveals high conservation in the HVR region.

With regards to the HAdV-F40 strains, two samples had a 100% nt identity between them, and showed the highest nt similarity (ranging from 99–100%) with F40 sequences isolated in Brazil in 2018 to 2019 (OQ442231 and OQ442228), with sequences from India (MT952560), and with the prototype F40 strain Dungan (L19443). Pairwise comparison of nt sequences revealed a range of 95.7–96.5% sequence identity between HAdV-F40 and HAdV-F41 strains.

Phylogenetic analysis of the shaft region in the fiber gene revealed two distinct lineages within HAdV-F41: F-GTC 1 and F-GTC 2 (Figure 5B). Among the analyzed samples, 23 were classified under the F-GTC 2 lineage, while three belonged to the F-GTC 1 lineage. A high nt similarity (99.8%) was observed between the two clusters of HAdV-F41 strains. Our Brazilian strains of the cluster F-GTC 1 samples showed the highest nt similarity (100%) with samples from Japan (AB610541 and AB610541), China (AB610538, MT150336, HM565136 and KY316160), India (MT952547), and Brazil (OQ442236, OQ442237, OQ442238 and OQ442239). Samples within the cluster F-GTC 2 showed high nt similarity (99.8–100%) with samples from Germany (ON532825 and ON815890), Sweden (KX868523), Brazil (OQ442259, OQ442255, and OQ442244), and China (MT150394). The identity between group F-GTC 1 and F-GTC 2 ranged from 98.1% to 99.2%

## 4. Discussion

Our study provides insights into the prevalence, epidemiological features, and genetic diversity of HAdV among AGE cases in medically attended patients (symptomatic children and adults) across nine Brazilian states during the COVID-19 pandemic (2021–2023). During the study period, we detected HAdV at a prevalence rate of 16.6%. HAdV-F41 was the most prevalent genotype and accounted for more than 38.2% of total infections. Although multiple HAdV species were detected, the enteric types demonstrated significantly higher viral DNA shedding compared to non-enteric types. We also identified the enteric genotype HAdV-F41 as the predominant strain, accounting for most of the identified cases.

Globally, the prevalence of HAdV varies significantly, ranging from 2% to 39%, with the highest detection rates observed in low- and lower-middle-income countries [54,55,56,57,58]. Limited access to clean water, inadequate sanitation infrastructure, and higher population density, usually observed in LMICs, are likely key factors contributing to discrepancies [59,60]. In Brazil, studies conducted up to 2017 reported a prevalence of HAdV among AGE cases ranging from 4% to 7% [36,58,61]. Between 2018 and 2020, Do Nascimento et al. [33] found an overall prevalence of HAdV of 24.5% in eleven Brazilian states. In another two studies conducted in Brazil, HAdV was detected at prevalence rates of 28.6% and 18.5% in the Southeastern and Amazon regions, respectively [62,63].

In our study, we observed a decline in HAdV circulation in Brazil during the pandemic period, especially in the first semester of 2021, when the measures to control the spread of SARS-CoV-2 were stringent. These measures included the enforcement of social distancing, lockdowns, childcare and school closures, and mandatory use of face masking. Previous HAdV surveillance conducted by our group reported similar trends during the year of 2020. In that study, HAdV prevalence rates were 26.7% and 27.9% in 2018 and 2019, respectively, and decreased to 12.6% in 2020. During this study, similar prevalence rates (13% and 15.9%) were found in 2021 and 2022. In 2023, HAdV prevalence rebounded to 20.1%, returning to levels comparable to those observed in the pre-pandemic years [33]. Since the beginning of the COVID-19 pandemic, several studies reported a decrease in the circulation of several respiratory and enteric pathogens, especially viral infections in the pediatric population [64,65,66,67,68]. For instance, studies conducted in Norway, Thailand, Germany, and Poland demonstrated a reduction in the circulation of respiratory viruses (such as influenza virus A/B and rhinovirus) and gastroenteric viruses (HAdV, norovirus, and rotavirus), varying from 2% to 87% during the pandemic years [69,70,71,72]. Additionally, our findings align with data from studies conducted in China, Iraq, Albania, and India, between 2011 to 2017, which detected HAdV in over 20% of tested stool samples from children with AGE [73,74,75,76]. Two studies have identified HAdV as the primary and secondary leading pathogen responsible for diarrhea requiring hospitalization in India and Bangladesh, respectively [77,78].

Regarding age groups, we observed the highest prevalence of HAdV infections in children aged up to 60 months old. Studies conducted across several countries in Africa, Asia and Latin America on bacterial and viral pathogens have shown that enteric HAdV exhibit a higher attributable incidence in children under two years of age compared to those up to 5 years old, with rates ranging from 3.9 to 20.9 episodes of diarrhea per 100 child-years [79,80,81].

The HAdV co-detection rate found in rotavirus- and norovirus-positive was 20%. Interestingly, only 5.5% of these co-detected cases involved the enteric types F40/41. A significantly higher viral load of rotavirus and norovirus was observed compared to HAdV, suggesting that those viruses were likely the primary causative agent of AGE symptoms, rather than HAdV. A higher co-detection rate was previously reported by our group in samples from Brazil during the 2018–2020 period [33]. Co-detection between different enteric viruses and HAdV is not an uncommon report. Two retrospective studies conducted in India among pediatric patients have shown that HAdV was the main agent found in co-detection with norovirus [82,83]. In Brazil, during a norovirus-associated AGE outbreak in the Santa Catarina state, southern region, in 2023, HAdV was co-detected in 6.8% of norovirus-positive samples [84]. Similarly, studies conducted in the United States and Canada have reported high rates of HAdV co-detection in AGE patients, with rates of 27% and 30.9%, respectively [85,86].

Overall, we found a higher predominance of enteric HAdV species (44.1%) compared to non-enteric species. Moreover, we demonstrated significantly higher viral DNA shedding in the group of F40/41-positive fecal samples. Given the ubiquitous nature of HAdV, with over 100 types capable of infecting humans and causing a wide range of clinical symptoms, including common conditions such as respiratory diseases and conjunctivitis, it is not surprising that non-enteric types were frequently detected [33,55,58,87,88,89,90].

During our study, we observed a significant predominance of HAdV-F41 (86.7%) compared to HAdV-F40 (13.3%) among the enteric types. Our data are similar to worldwide findings, revealing a global trend regarding HAdV genotype distribution among AGE cases [27,54,55,85,86]. In Brazil, during a surveillance study conducted in the same regions between 2018 and 2020, Nascimento et al., 2022 [33] observed that among HAdV-F species, type F41 was the most prevalent genotype (81.5%) [33]. This genotype was significantly more common than others, particularly among children under 24 months of age. In line with our data, other studies conducted in Brazil have reported similar findings, revealing that over 50% of the sequenced samples belonged to the HAdV-F41 genotype [35,36,37,58,91]. In Australia, a study using next-generation sequencing identified a pronounced predominance of serotype F41 (83.5%) among six major human adenovirus (HAdV) groups isolated from wastewater samples in Sydney and Melbourne, between 2016 and 2017 [92]. In this same study, clinical data further confirmed F41 as the predominant serotype, accounting for 52.5% of gastroenteritis cases, with C1 and C2 exhibiting lower prevalence rates [89]. In Venezuela, during a 13-year surveillance study conducted before and after the implementation of a rotavirus vaccine, HAdV was detected in 19.5% of AGE samples, with the domain (79.8%) of the genotype F41 [93]. More recently, a study conducted in India during the year of 2019 identified HAdV-F as the leading cause of AGE in children under 5 years of age, with a prevalence rate of 49.4% [94].

Phylogenetic analysis of the HVR region revealed that most Brazilian F41 strains grouped into the H-GTC1 cluster (*n* = 22), with a smaller proportion belonging to the H-GTC 2 lineage (*n* = 6). Only one sample was identified as HAdV-F40. These findings contrast with those of a previous study by our group, where the majority of the samples were classified as H-GTC 2 [42]. Studies carried out in Germany and India have also reported the co-circulation of multiple HAdV-F41 lineages, with lineage 2 being particularly prevalent [94,95]. Interestingly, over the six-year surveillance period, our group observed a shift in the circulation of these lineages in Brazil. The reason for this remains unclear, but the data suggest that immune pressure may have driven the genotype replacement.

From the shaft region, we observed a predominance of cluster F41-GTC 2 (*n* = 23) compared to F41-GTC 1 (*n* = 3). Only two samples were detected as HAdV-F40. These data are similar to those observed in Kenya, Brazil, and India [30,42,94]. Recent studies analyzing the complete genome of HAdV-F41 have identified the hexon and fiber (both short and long) genes, as well as the E3 and E4 regions, as the main sites of mutation within the HAdV genome [30,31,95].

During our study, among the non-enteric HAdV types, the most prevalent species was HAdV-C (52.6%), represented by genotypes C1 and C2, followed by HAdV-B3. Several studies have reported that non-enteric HAdV genotypes, frequently detected in AGE stool samples, primarily belong to species A, B, C, and D, with HAdV-C being the most common. However, these genotypes are typically associated with other diseases rather than AGE [20,96,97,98].

Our study has some limitations. Firstly, we did not access detailed clinical/epidemiological data from patients, which hindered our ability to track non-enteric HAdV cases with other possible clinical symptoms. Secondly, we did not investigate non-viral enteropathogens, such as bacteria and parasites, or other AGE-related viruses that may be involved in clinical cases of AGE. Moreover, our study did not include an asymptomatic group, which could help us to consider HAdV’s involvement in clinical symptoms. Thirdly, it was not possible to sequence all HAdV-positive samples, given that many samples showed high Ct values (Ct > 32), indicating lower viral loads. Our evolutionary analysis of species F was based on partial parts of the hexon and long fiber genes rather than the complete genome. Finally, we believe that a portion of HAdV-positive samples may represent prolonged viral shedding after previous infections and may not contribute to AGE in all positive cases, especially in samples with co-viral findings, non-enteric types, and low viral loads.

## 5. Conclusions

In conclusion, we describe here the epidemiological and molecular features of HAdV in medically attended patients with AGE in Brazil. We assessed the viral shedding, age distribution, and genetic diversity of HAdV-F40/41 strains. After the COVID-19 pandemic period, from 2021 to 2023, we identified HAdV in 16.6% of samples. The surveillance and molecular epidemiology of HAdV are essential to assess the circulation patterns of both enteric and non-enteric types, as well as to identify potential emerging strains at the population level. Our data demonstrate that HAdV is an important cause of AGE and should be included in diagnostic protocols along with rotavirus and norovirus, which are routinely tested in AGE surveillance programs in Brazil and many other countries. Therefore, maintaining expanded national surveillance is crucial to support public health and monitor future trends and changes in HAdV epidemiology in Brazil.

## Figures and Tables

**Figure 1 viruses-17-00577-f001:**
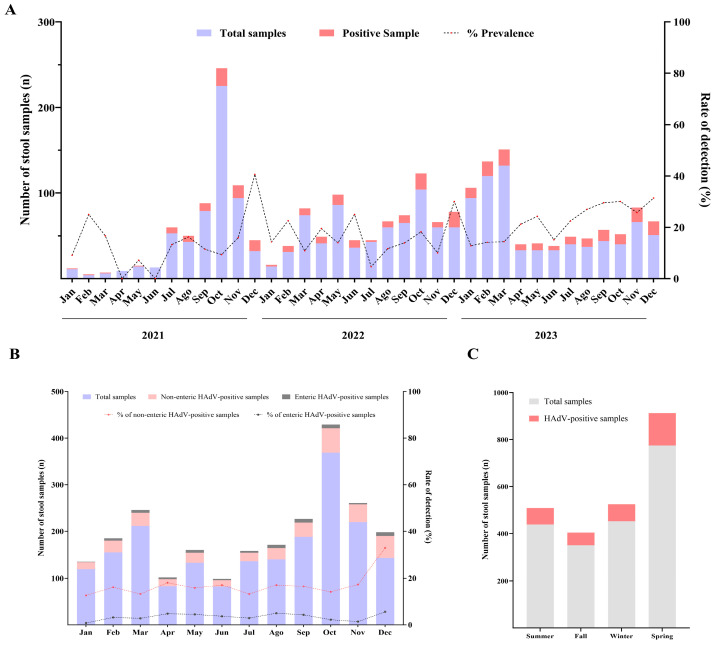
(**A**) Monthly distribution of tested acute gastroenteritis samples, HAdV-positive samples, and detection rates in Brazil, 2021–2023. (**B**) Monthly distribution of surveillance period addition of acute gastroenteritis samples tested. HAdV-positive samples and detection rates in Brazil. (**C**) Seasonal distribution period addition of HAdV-positive samples for each Brazilian region.

**Figure 2 viruses-17-00577-f002:**
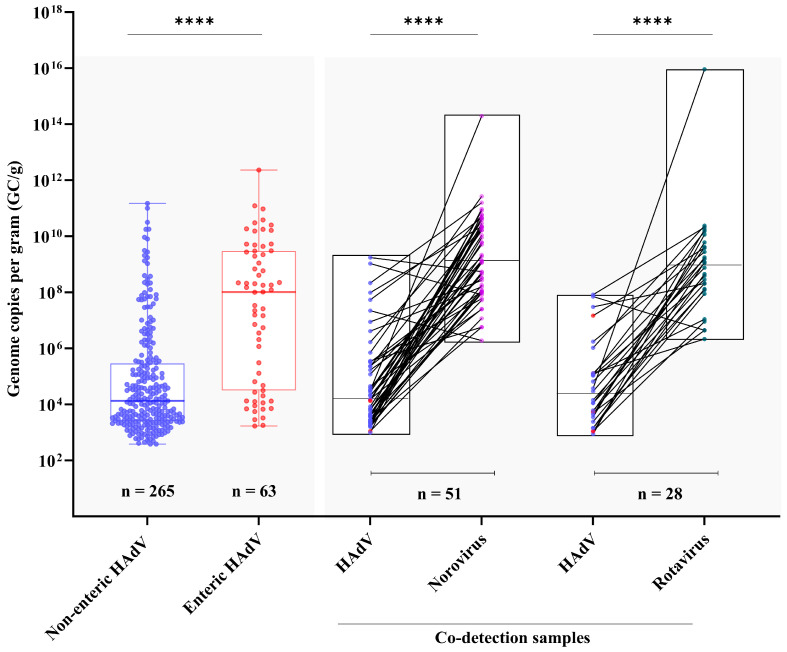
Distribution of viral load of non-enteric and enteric HAdV, noroviruses, and rotaviruses in stool samples, expressed as genome copies per gram (GC/g). Box-and-whisker plots show the first and third quartiles (equivalent to the 5th and 95th percentiles), the median (the horizontal line in the box). **** *p* ≤ 0.0001.

**Figure 3 viruses-17-00577-f003:**
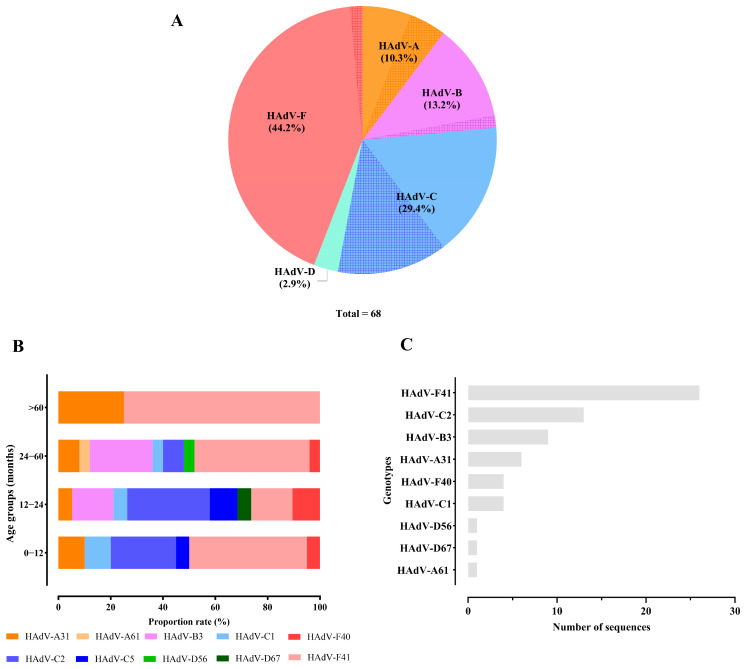
(**A**) The total frequency of HAdV types obtained from samples sequenced between 2021 and 2023. HAdV types obtained from co-detection with rotavirus or norovirus are represented by a hatched pattern. (**B**) Frequency by age group. (**C**) Distribution of HAdV genotypes identified in samples from Brazil between 2021 and 2023.

**Figure 4 viruses-17-00577-f004:**
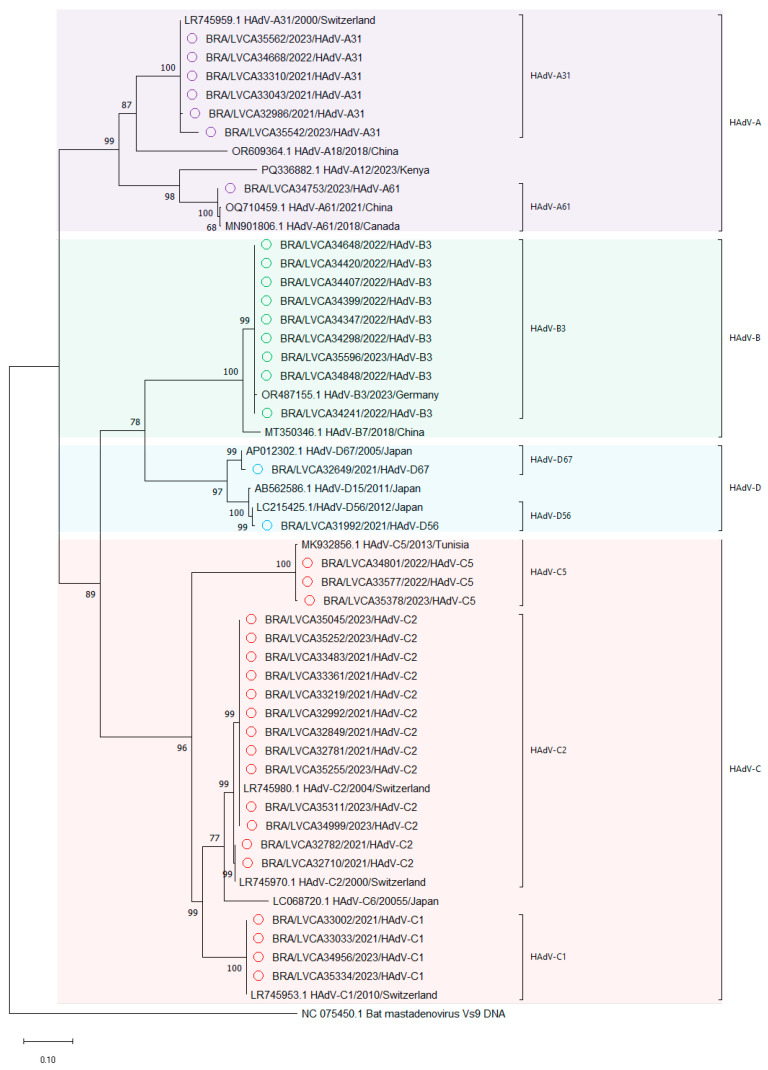
Phylogenetic tree based of non-enteric HAdV on the conserved region of the nucleotide (nt) sequences of the hexon gene. Strains isolated in this study (marked with a circle) are shown as country, LVCA internal register number, year of collection, and type (i.e., BRA/LVCA35334/2023/HAdV-C1) and Reference strains were downloaded from GenBank. Maximum likelihood phylogenetic trees were constructed using MEGA X software with bootstrap testing (1000 replicates) applied the on Tamura-Nei (TN93) +G + I. Bootstrap values above 60% are given at branch nodes.

**Figure 5 viruses-17-00577-f005:**
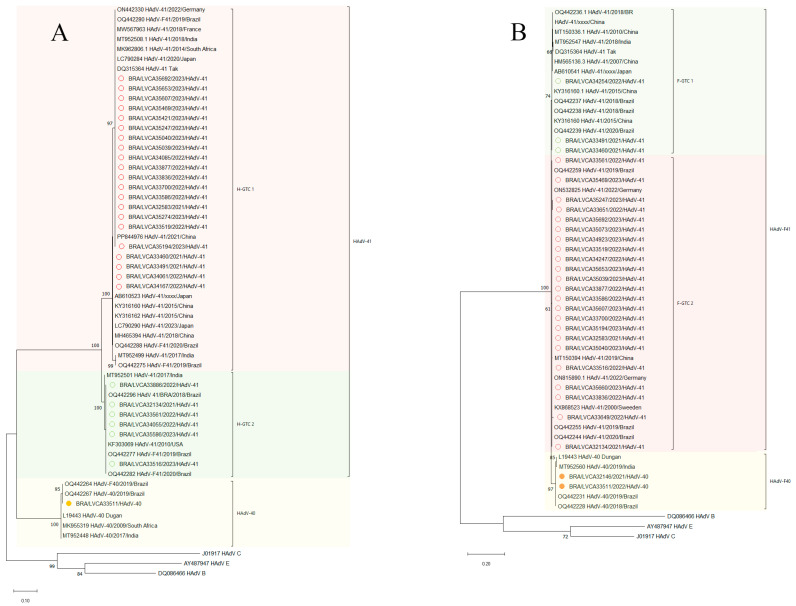
Phylogenetic tree based on the nt sequence of the (**A**) hypervariable regions (HVR1–HVR6) of the hexon gene and (**B**) partial shaft region of the fiber gene. HAdV-F40 and -F41. Strains isolated in this study (marked with a circle) are shown as country, LVCA internal register number, year of collection, and type (i.e., BRA/LVCA33516/2023/HAdV-41) and Reference strains were downloaded from GenBank. Maximum likelihood phylogenetic trees were constructed using MEGA X software with bootstrap testing (1000 replicates), applying the HKY model for the hexon hypervariable regions (**A**) and the HKY + I model for the fiber shaft region (**B**). Bootstrap values ≥ 60% are shown at branch nodes.

**Table 1 viruses-17-00577-t001:** Number of tested and HAdV-positive stool samples identified through laboratory-based surveillance by year and states in Brazil from 2021 to 2023.

States	No. Stool Samples—Positive/Total (%)			
	2021	2022	2023	Total
Bahia	8/55 (14.5)	0/2	0/0	8/57 (14)
Paraíba	0/0	1/2	10/28 (35.7)	11/30 (36.7)
Pernambuco	5/9	14/100 (14)	18/94 (19.1)	37/203 (18.2)
Sergipe	0/1	2/2	3/6	5/9
Espírito Santo	3/22 (13.6)	5/87 (5.7)	0/5	8/114 (7)
Minas Gerais	4/41(9.8)	14/81 (17.3)	66/230 (28.7)	84/352 (23.9)
Rio de Janeiro	0/18	6/39 (15.4)	1/4	7/61 (11.5)
Rio Grande do Sul	33/332 (10)	44/241 (18.3)	33/125 (26.4)	110/698 (15.8)
Santa Catarina	23/105 (21.9)	21/120 (17.5)	14/231 (6)	58/456 (12.7)
Total	76/583 (13)	107/674 (15.9)	145/723 (20.1)	328/1980 (16.6)

**Table 2 viruses-17-00577-t002:** Distribution of tested and HAdV-positive stool samples identified through laboratory-based surveillance by region in Brazil from 2021 to 2023.

Region	No. of Stool Samples—Positive/Total (%)			*p*-Value ^a^ (Fisher Test)			
	2021	2022	2023	Total	2021 vs. 2022	2021 vs. 2023	2022 vs. 2023
Northeastern	13/65 (20)	17/106 (16)	31/128 (24.2)	61/299 (20.4)	0.5387	0.5880	0.1443
Southeastern	7/81 (8.6)	25/207 (12.1)	67/239 (28)	99/527 (18.8)	0.5323	0.0002	<0.0001
Southern	56/437 (12.8)	65/361 (18)	47/356 (13.2)	168/1154 (14.6)	0.0473	0.9156	0.0810
Total	76/583 (13)	107/674 (15.9)	145/723 (20.1)	328/1980 (16.6)	0.1728	0.0008	0.0437

^a^ The *p*-values are indicated on the table (Fisher test).

**Table 3 viruses-17-00577-t003:** Number of tested and HAdV-positive fecal samples by age group in Brazil during 2021–2023.

Age Groups ^a^	No. of Fecal Samples—Positive/Tested (%)			Total		*p*-Value ^b^
	2021	2022	2023	All HAdV	HAdV-F	
0–12	16/91 (17.6)	22/101 (21.8)	39/156 (25)	77/348 (22.1)	14/348 (4)	<0.0001
12–24	20/110 (18.2)	29/132 (22)	44/135 (32.6)	93/377 (24.7)	17/377 (4.5)	<0.0001
24–60	25/133 (18.8)	41/157 (26.1)	35/142 (24.7)	101/432 (23.4)	23/432 (5.3)	<0.0001
>60	15/249 (6)	15/284 (5.3)	27/290 (9.3)	57/823 (6.9)	9/823 (1.1)	-

^a^ age in months; ^b^ *p*-value < 0.0001 compared to all < 60 months. The *p*-values are indicated in the table (Fisher test).

## Data Availability

The data that support the findings of this study are openly available in the GenBank database. The datasets generated and analyzed during the current study are available in the GenBank repository under the following accession numbers cited in the Materials and Methods. This study is registered in the Brazilian National System for Genetic Heritage and Associated Traditional Knowledge Management (No. A837EB6).

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
