# Peer review of "Molecular Epidemiology of Human Adenovirus from Acute Gastroenteritis Cases in Brazil After the COVID-19 Pandemic Period, 2021–2023"

_viruses, 2025, doi:10.3390/v17040577_

Round 1
Reviewer 1 Report
Comments and Suggestions for Authors
This study is well presented. Only minor changes are needed.
- In the study the authors indicated that “a total of 1980 stool samples collected from ten states were analyzed, revealing the presence of HAdV in 16.6% of cases, with the highest prevalence observed in children under five years of age. Among these, enteric HAdV (HAdV-F40/41) was detected in 3.2% of cases … ”. “Among these” is not clear . By reading the manuscript, we could know that the 3.2% is based on the 1980 stool samples, not among the HAdV positive cases. Please clarify this. In addition, it will be better if the numbers of positive cases for 16.6% and 3.2% are added in the abstract, like in line 187-188.
- In line 28-30, “… in low- and middle-income countries (LMICs) [1,2]. It accounts for approximately 1.7 billion cases and 1.1 million deaths annually,” You may add the word “worldwide” in here, if the cases and deaths are not limited in low- and middle-income countries.
- This study covers 10 states as indicated by the authors in abstract and in line 81, but only 9 state are listed in table 1. Why?
- In line 227-228, as well as the other text, “The viral loads found in the samples ranged from 1.7 × 102 CG/g (genomic copies per 227 gram of feces) to 2.3 × 1012 GC/g.” GC or CG?
Author Response
Comments 1: In the study the authors indicated that “a total of 1980 stool samples collected from ten states were analyzed, revealing the presence of HAdV in 16.6% of cases, with the highest prevalence observed in children under five years of age. Among these, enteric HAdV (HAdV-F40/41) was detected in 3.2% of cases … ”. “Among these” is not clear . By reading the manuscript, we could know that the 3.2% is based on the 1980 stool samples, not among the HAdV positive cases. Please clarify this. In addition, it will be better if the numbers of positive cases for 16.6% and 3.2% are added in the abstract, like in line 187-188.
Response 1: As suggested, we revised the abstract to enhance clarity regarding HAdV detection rates by including the absolute numbers of positive cases corresponding to the reported percentages.
Comments 2: In line 28-30, “… in low- and middle-income countries (LMICs) [1,2]. It accounts for approximately 1.7 billion cases and 1.1 million deaths annually,” You may add the word “worldwide” in here, if the cases and deaths are not limited in low- and middle-income countries.
Response: Done
Comments 3: This study covers 10 states as indicated by the authors in abstract and in line 81, but only 9 state are listed in table 1. Why?
Response 3: In fact, we received sample from nine states. This information has been corrected throughout the text.
Comments 4: In line 227-228, as well as the other text, “The viral loads found in the samples ranged from 1.7 × 102 CG/g (genomic copies per 227 gram of feces) to 2.3 × 1012 GC/g.” GC or CG?
Response 4: The correct abbreviation is genome copies per gram (GC/g). This error has been corrected throughout the manuscript.
Reviewer 2 Report
Comments and Suggestions for Authors
Review
The authors assessed the molecular epidemiology of AdV-F40/F41 in AGE cases reported between 2021 to 2023 after the COVID-19 period.
The screening of 1980 samples revealed 16.6 % of HAdV positive with HAdV-F41 as the predominant genotype. The identified strains from the study were closely related to worldwide reference strains suggesting potential role of HAdV associated-AGE in Brazil.
- Title
I suggest the title as follows: “Molecular epidemiology of Human Adenovirus from acute gastroenteritis cases in Brazil after the COVID-19 pandemic period, 2021-2023”.
- Abstract
-You need to specify if the samples collected were from outpatients or hospitalised.
-The methodology is missing. How did you proceed to achieve technically your objectives?
- Introduction
-Lines 53-54: As you know the author, it will be clear to state simply “Cohen et al. who worked in LMICs …”.
-Line 55: -The study quoted in reference 29 is not recent study. It has been reported in 2021!
- Correct: The study …in children younger than six years old has been identified ...
-Line 66: -Correct: … initial vaccination.
- Materials and Methods
-Lines 84-85: Were the stool samples collected from outpatients or hospitalised individuals?
-Line 132: Correct: The conventional PCR was performed for a conserved region…
- Results
-Lines 203,215 and 222.
The titles for table1, 2 and 3 are not correct. They should be revised accordingly with the table contents.
-Figure 3: Why the authors set the groups: > 12-24 and > 24-60? Is it not 12-24 and 24-60?
-Lines 272-282: Which application was used to calculate the nucleotides percent similarity?
-Phylogenetic analysis: The number of references used to set different trees are not mentioned. On the basis of which criteria were they selected?
-Lines 356-357: The sentence can be rephrased by stating the main author for example:
Do Nascimento et al. 2022 reported…
-Lines 364-377: This comparison of HAdV prevalence rates is missing enough data from LMICs as that could confirm the authors statement given in the second paragraph of the discussion (lines 351-354).
-Line 426: List the other countries you want to refer to.
-Line 433: Name respectively the other countries you are referring to in this sentence.
-Lines 449-453: Limitation: The screening of HAdV in the asymptomatic group could help as well to consider or not it involvement in symptomatic cases.
Line 457: Correct: After the COVID-19…
Author Response
Comments 1: The authors assessed the molecular epidemiology of AdV-F40/F41 in AGE cases reported between 2021 to 2023 after the COVID-19 period.
The screening of 1980 samples revealed 16.6 % of HAdV positive with HAdV-F41 as the predominant genotype. The identified strains from the study were closely related to worldwide reference strains suggesting potential role of HAdV associated-AGE in Brazil.
Response 1: The authors would like to thank the reviewer for his contribution to improve the quality of the manuscript.
Comments 2: Title
I suggest the title as follows: “Molecular epidemiology of Human Adenovirus from acute gastroenteritis cases in Brazil after the COVID-19 pandemic period, 2021-2023”.
Response 2: The title was changed according to this reviewer’s suggestion.
Comments 3: Abstract
Comments 3a: -You need to specify if the samples collected were from outpatients or hospitalised.
Response 3a: Stool samples were collected from medically attended patients with symptoms of acute gastroenteritis (AGE), both children and adults. The information was modified within the manuscript (abstract and M&M) to make it clear.
Comments 3b: -The methodology is missing. How did you proceed to achieve technically your objectives?
Response 3b: We included methodology in the abstract as suggested.
Comments 4: Introduction
Comments 4a: -Lines 53-54: As you know the author, it will be clear to state simply “Cohen et al. who worked in LMICs …”.
Response 4a: Done.
Comments 4b: -Line 55: -The study quoted in reference 29 is not recent study. It has been reported in 2021!
- Correct: The study …in children younger than six years old has been identified ...
Response 4b: Done.
Comments 4c: -Line 66: -Correct: … initial vaccination.
Response 4c: Done.
Comments 5: Materials and Methods
Comments 5a: -Lines 84-85: Were the stool samples collected from outpatients or hospitalised individuals?
Response 5a: Stool samples were collected from medically attended patients with symptoms of acute gastroenteritis (AGE), both children and adults.
Comments 5b: -Line 132: Correct: The conventional PCR was performed for a conserved region…
Response 5b: Done.
Comments 6: Results
Comments 6a: -Lines 203,215 and 222.
The titles for table1, 2 and 3 are not correct. They should be revised accordingly with the table contents.
Response 6a: Done
Comments 6b: -Figure 3: Why the authors set the groups: > 12-24 and > 24-60? Is it not 12-24 and 24-60?
Response 6b: We revised Figure 3, as suggested, removing the “>” from the groups.
Comments 6c: -Lines 272-282: Which application was used to calculate the nucleotides percent similarity?
Response 6c: Phylogenetic analysis and nucleotides percent similarities were performed using the MEGA X software, as indicated in the M&M.
Comments 6d: -Phylogenetic analysis: The number of references used to set different trees are not mentioned. On the basis of which criteria were they selected?
Response 6d: We used the prototypes strains (Dugan and Tak) for HAdV 40 and 41, respectively. Additionally, we used HAdV sequences for each cluster H-GTC 1 and 2, similarly to our previous study (Nascimento et al., 2023), that was based on an HAdV phylogenetic study that originally described the used primers for each region (Xu et al., 2000; Li et al., 2004). We also selected some HAdV sequences based on their high homology by using BLAST.
Comments 6e: -Lines 356-357: The sentence can be rephrased by stating the main author for example:
Do Nascimento et al. 2022 reported…
Response 6e: Done
Comments 6f: -Lines 364-377: This comparison of HAdV prevalence rates is missing enough data from LMICs as that could confirm the authors statement given in the second paragraph of the discussion (lines 351-354).
Response 6f: We included new data supporting HAdV as a leading cause of AGE in LMICs, as suggested.
Comments 6g: -Line 426: List the other countries you want to refer to.
Response 6g: Done.
Comments 6h: -Line 433: Name respectively the other countries you are referring to in this sentence.
Response 6h: Done.
Comments 6i: -Lines 449-453: Limitation: The screening of HAdV in the asymptomatic group could help as well to consider or not it involvement in symptomatic cases.
Response 6i: We included this information in the limitation paragraph in the end of the Discussion section.
Comments 6j: Line 457: Correct: After the COVID-19…
Response 6j: Done.